# Anti-Spike Antibody Response to Natural Infection with SARS-CoV-2 and Its Activity against Emerging Variants

Cheng-Pin Chen,[a,b] Kuan-Ying A. Huang,[c,d,i] Shin-Ru Shih,[c,e,i] Yi-Chun Lin,[a,f] Chien-Yu Cheng,[a,g] Yhu-Chering Huang,[d] Tzou-Yien Lin,[d] Shu-Hsing Cheng[a,h]

aDepartment of Infectious Diseases, Taoyuan General Hospital, Ministry of Health and Welfare, Taoyuan, Taiwan
bInstitute of Clinical Medicine, National Yang Ming Chiao Tung University, Taipei, Taiwan
cResearch Center for Emerging Viral Infections, College of Medicine, Chang Gung University, Taoyuan, Taiwan
dGenomics Research Center, Academia Sinica, Taipei, Taiwan
eDepartment of Laboratory Medicine, Chang Gung Memorial Hospital, Taoyuan, Taiwan
fGraduate Institute of Clinical Medicine, College of Medicine, Taipei Medical University, Taipei, Taiwan
gInstitute of Public Health, National Yang Ming Chiao Tung University, Taipei, Taiwan
hSchool of Public Health, Taipei Medical University, Taipei, Taiwan
iDivision of Pediatric Infectious Diseases, Department of Pediatrics, Chang Gung Memorial Hospital, Taoyuan, Taiwan

Cheng-Pin Chen and Kuan-Ying A. Huang contributed equally to this article. Author order was determined in the order of increasing seniority.

**ABSTRACT** The outbreak of severe acute respiratory syndrome coronavirus 2 (SARS-CoV-2) has substantially affected human health globally. Spike-specific antibody response plays a major role in protection against SARS-CoV-2 infection. Here, we examined serological anti-spike antibody and memory B cell responses in adults with acute SARS-CoV-2 infection. Twenty-five adult patients were enrolled between January and September 2020, and 21 (84%) had a detectable spike-binding antibody response in serum on day $21 \pm 8$ (6 to 33) after the onset of illness. Among those with positive spike-binding antibody response, 19 (90%) had a positive hemagglutination titer and 15 (71%) had angiotensin-converting enzyme 2 (ACE2)-blocking serological activities. Follow-up serum samples collected $11 \pm 1$ (7 to 15) months after infection exhibited an average of $2.6 \pm 1.0$ (1.0 to 3.5)-fold reduction in the spike-binding antibody response. Moreover, convalescent and follow-up serum samples showed $83 \pm 82$ (15 to 306)- and $165 \pm 167$ (12 to 456)-fold reductions in the neutralization activity against the Omicron variant, respectively. Upon acute infection, spike-specific memory B cell responses were elicited, with an average frequency of $1.3\% \pm 1.2\%$ of peripheral B cells on day $19 \pm 7$ (6 to 33) after the onset of illness. IgM memory B cells were predominantly induced. Patients with fever and pneumonia showed significantly stronger spike-binding, ACE2-blocking antibody, and memory B cell responses. In conclusion, spike-specific antibody response elicited upon acute SARS-CoV-2 infection may wane over time and be compromised by the emergence of viral variants.

**IMPORTANCE** As spike protein-specific antibody responses play a major role in protection against SARS-CoV-2, we examined spike-binding and ACE2-blocking antibody responses in SARS-CoV-2 infection at different time points. We found robust responses following acute infection, which waned approximately 11 months after infection. Patients with fever and pneumonia showed significantly stronger spike-binding, ACE2-blocking antibody, and memory B cell responses. In particular, spike-specific antibody response in the convalescent and follow-up serum samples was substantially affected by emerging variants, especially Beta and Omicron variants. These results warrant continued surveillance of spike-specific antibody responses to natural infections and highlight the importance of maintaining functional anti-spike antibodies through immunization.

**KEYWORDS** COVID-19, neutralization antibody, memory B cell, Omicron, SARS-CoV-2

Address correspondence to Shu-Hsing Cheng, shcheng@mail.tygh.gov.tw.

The authors declare no conflict of interest.

Novel coronavirus disease (COVID-19) was first reported at the end of 2019 (1). It spread rapidly and was declared a global health emergency (1, 2). As of February 2022, nearly 390 million confirmed cases of COVID-19 and over 5 million deaths have been reported to the World Health Organization (3). The causative agent of COVID-19 is severe acute respiratory syndrome coronavirus 2 (SARS-CoV-2) (4).

SARS-CoV-2 is an enveloped betacoronavirus with protrusions of large trimeric "spike" (S) proteins. Receptor binding domains (RBDs) located at the tip of these spikes facilitate host cell entry via interaction with human angiotensin-converting enzyme 2 (ACE2) (5, 6). After entry, the SARS-CoV-2 nucleocapsid internalizes into the host cell and uses the host ribosome to produce its own mRNA, which then continuously synthesizes viral proteins in the cell cytoplasm, resulting the construction of new viral particles (7). SARS-CoV-2 causes a broad spectrum of clinical manifestations, ranging from asymptomatic infection to mild to moderate disease, including upper and lower respiratory symptoms, to critical illness requiring intubation and intensive care (1, 8, 9); in addition, it elicits a complex immune response (10).

Since the beginning of the pandemic, the detection and measurement of nucleoprotein (N), spike, and receptor-binding domain antibodies against SARS-CoV-2 have been used to determine SARS-CoV-2 infection, outbreak investigation, seroprevalence (11, 12), and vaccine efficacy and coverage (13, 14). Furthermore, accumulating evidence has shown the importance of antibody-mediated immunity against SARS-CoV-2 infection and the development of severe illnesses following infection in humans (15, 16).

As SARS-CoV-2 continues to spread and cause outbreaks worldwide, understanding the immune response to SARS-CoV-2 is increasingly essential. Therefore, in this study, we focused on the magnitude, function, and longevity of the anti-spike antibody response to natural infection in humans. Variants of SARS-CoV-2 have been reported in many countries worldwide and harbor critical mutations in the spike protein. Therefore, we explored the effect of emerging variants on anti-spike antibodies elicited by natural infections in adults.

## RESULTS

**Clinical manifestations and spectrum of natural SARS-CoV-2 infection.** In total, 25 adult patients with acute SARS-CoV-2 infection were enrolled. Their mean age was 38.68 (standard deviation [SD], 13.4) years, and the male-to-female ratio was 12:13. All patients (23 Taiwanese adults and 2 foreign adults) were identified as imported cases of COVID-19 by Taiwan Centers for Disease Control. Among the enrolled patients, five developed pneumonia, as confirmed by chest radiography or computed tomography. The mean age of patients with and without pneumonia was 47.4 and 36.5 years, respectively ($P = 0.11$). Fever (60%) and cough (60%) were frequently reported symptoms. Patients with pneumonia had a higher level of alanine aminotransferase (ALT) (46.8 U/L versus 26.6 U/L; $P = 0.02$) than those without pneumonia. Patients with and without pneumonia did not have remarkable differences in white blood cell count, C-reactive protein, creatinine, and lactate dehydrogenase and received similar therapeutic regimens (Table 1).

**Anti-spike antibody response.** A total of 48 convalescent-phase serum samples were collected from 25 COVID-19 patients on days 5 to 39 after onset of symptoms. For each patient, at least one serum sample was collected, and 14 patients provided additional convalescent-phase serum samples during hospitalization (see Table S1 in the supplemental material).

Twenty-one (84%) patients presented a detectable spike-binding antibody response in the serum on day $21 \pm 8$ (6 to 33) after the onset of illness (Fig. 1A; see also Table S2 in the supplemental material). Four patients (patients 3, 5, 8, and 9) were negative for the anti-spike antibody response; of these patients, all presented mild illness and three of them did not have fever (Fig. 1A; Tables S1 and S2).

The spike-binding antibody response was detected as early as the first week after illness onset (binding activity, $17\% \pm 8\%$; $n = 5$), continued to rise between the second (binding activity, $39\% \pm 8\%$; $n = 11$) and third weeks (binding activity, $51\% \pm 7\%$; $n = 12$) (analysis of variance [ANOVA] with *post hoc*, $P < 0.05$), and plateaued 4 weeks after onset (binding activity, $52\% \pm 4\%$; $n = 20$) (ANOVA with *post hoc*, $P < 0.01$) during the convalescence period

**TABLE 1** Demographics and clinical description of 25 COVID-19 cases

| Demographic | No. (%) of patients[a] | | | |
| --- | --- | --- | --- | --- |
| | Pneumonia | No pneumonia | Total | P value |
| All patients | 5 (20) | 20 (80) | 25 (100) | |
| Mean (± SD) age | 47.4 (10.5) | 36.5 (13.4) | 38.68 (13.4) | 0.105 |
| Male | 3 (60) | 9 (45) | 12 (48) | 0.459 |
| Female | 2 (40) | 11 (55) | 13 (52) | |
| Imported cases | 4 (80) | 17 (75) | 21 (84) | 0.720 |
| Europe | 0 | 6 (30) | 6 (24) | |
| America | 1 (20) | 3 (15) | 4 (16) | |
| Africa | 1 (20) | 2 (10) | 3 (12) | |
| Asia-pacific | 2 (40) | 6 (30) | 8 (32) | |
| Indigenous cases | 1 (20) | 3 (15) | 4 (16) | |
| Occupation | | | | 0.720 |
| Student | 0 | 5 (25) | 5 (20) | |
| Military | 1 (20) | 2 (10) | 3 (12) | |
| Business | 1 (20) | 5 (25) | 6 (24) | |
| Professional | 2 (40) | 6 (30) | 8 (32) | |
| Other | 1 (20) | 2 (10) | 3 (12) | |
| Subject symptoms | | | | |
| Fever | 3 (60) | 9 (45) | 12 (48) | 0.459 |
| Cough | 3 (60) | 9 (45) | 12 (48) | 0.459 |
| Diarrhea | 1 (20) | 6 (30) | 7 (28) | 0.564 |
| Rhinorrhea | 0 | 7 (35) | 7 (28) | 0.161 |
| Sore throat | 0 | 6 (30) | 6 (24) | 0.219 |
| General soreness | 2 (40) | 4 (20) | 6 (24) | 0.343 |
| Abnormal smell | 0 | 6 (30) | 6 (24) | 0.219 |
| Abnormal taste | 0 | 5 (25) | 5 (20) | 0.292 |
| Avg no. (SD) of symptoms | 2.4 (0.54) | 3.1 (2.28) | 2.9 (2.06) | 0.539 |
| Laboratory data | | | | |
| WBC $10^9$/L mean (SD) | 6,252 (2,653) | 5,834 (1,880) | 5,963 (1,962) | 0.865 |
| Lymphocytes $10^9$/L mean (SD) | 1,211 (616) | 1,511 (628) | 1,481 (630) | 0.347 |
| CRP mg/dL mean (SD) | 1.16 (0.91) | 0.57 (0.72) | 0.67 (0.78) | 0.143 |
| Creatinine mg/dL mean (SD) | 0.96 (0.29) | 0.76 (0.13) | 0.80 (0.18) | 0.194 |
| ALT U/L mean (SD) | 46.8 (22.8) | 26.6 (13.9) | 30.6 (17.5) | 0.018 |
| LDH U/L mean (SD) | 258.8 (60.3) | 214.2 (42.4) | 224.4 (49.3) | 0.074 |
| Antiviral agent[b] | 4 (80) | 7 (35) | 11 (44) | 0.096 |
| Immune modulator[c] | 4 (80) | 13 (65) | 17 (68) | 0.475 |
| Steroid | 2 (40) | 1 (5) | 3 (12) | 0.091 |

[a]Data are presented as number of patients (%) unless otherwise indicated.
[b]Among pneumonia cases, 2 had lopinavir, 1 had both lopinavir and remdesivir, and 1 had oseltamivir. Among nonpneumonia cases, 6 had oseltamivir, and 1 had lopinavir.
[c]Hydroxychloroquine had been used in 4 pneumonia cases and 12 nonpneumonia cases. Colchicine had been used in 1 pneumonia case and 1 nonpneumonia case.

(Fig. 1B to D). In most cases, the RBD-binding antibody response was detectable at the same time point, whereas the spike-binding antibodies were elicited (Fig. 1; Table S2).

We also examined the level of functional anti-spike antibodies in the serum using hemagglutination and ACE2-blocking assays. Among 21 patients with positive spike-binding antibody response, 19 (90%) showed detectable hemagglutination titers, and 15 (71%) showed ACE2-blocking serological activities (Fig. 1A; Table S2).

The magnitude of the spike-binding antibody response was significantly correlated with RBD-binding, hemagglutination, and the ACE2-blocking antibody response in patients with COVID-19 (see Fig. S1 in the supplemental material), indicating the protective potential of anti-spike antibodies elicited upon natural infection.

**B cell response.** The development of memory B cell responses to spike protein constitutes a major part of the humoral immune memory against SARS-CoV-2. Spike-specific memory B cell responses to natural infection were measured using enzyme-linked immunosorbent spot (ELISpot) assay (Fig. 2A). Spike-specific memory B cell responses were elicited, with an average

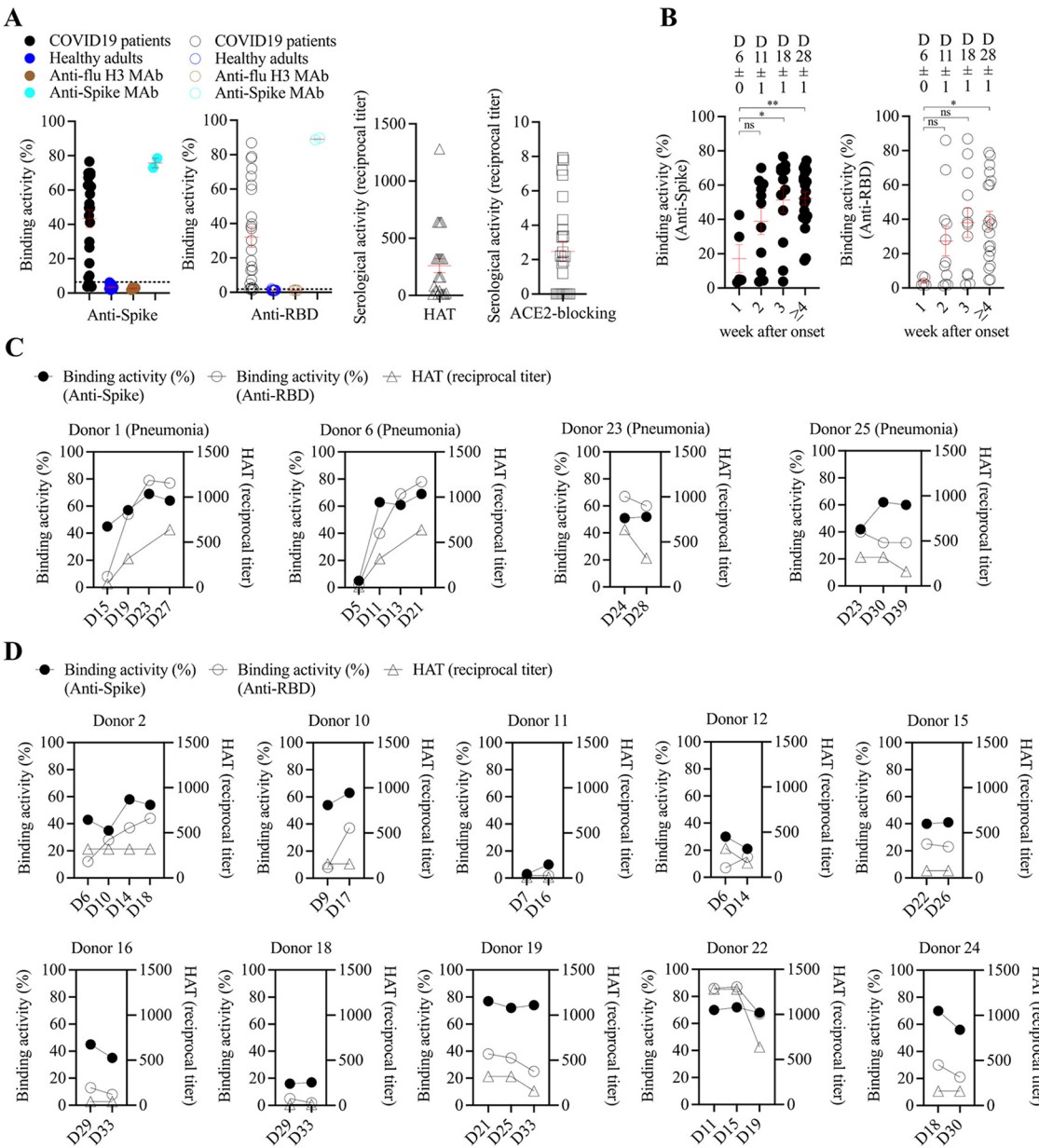

**FIG 1** Anti-spike antibody response to natural infection with SARS-CoV-2. (A) Spike-binding, RBD-binding, hemagglutination, and ACE2-blocking activity of serum samples from COVID-19 adult patients ($n = 25$). Each point represents peak serological activity for each patient, and the red line represents mean ± standard error of mean. Spike- and RBD-binding activities were measured by flow cytometry-based binding assay, and the activity was determined by the percentage of antibody-bound spike (or RBD)-expressing cells of serum minus the percentage of antibody-bound cells of PBS control. Healthy adult serum samples collected in 2017 were included as serum control. Spike (or RBD)-binding activity that was above two standard deviations plus the mean of healthy adult serum samples (dashed line) was positive. Anti-flu H3 and anti-SARS-CoV-2 spike human monoclonal antibodies were included as controls in the binding assay. For hemagglutination (HAT) assay, a serological titer of 1:20 or more was considered to be positive. Samples with a reciprocal titer of less than 20 were assigned a value of 10. The ACE2-blocking titer was expressed as the reciprocal of the serum dilution, giving 50% inhibition of signal compared to maximum signal. The original, undiluted serum that failed to inhibit ACE2-RBD interaction was scored as negative. (B) Spike- and RBD-binding serological activities during the course of COVID-19 in 25 patients. There were 5, 11, 12, and 20 serum samples collected in the 1st, 2nd, 3rd, and over the 4th week of illness, respectively. The mean ± standard error of mean of days after onset for each group was indicated. The significance between serological activities at different time points was determined using one-way ANOVA with *post hoc* Tukey's test. *, $P < 0.05$; **, $P < 0.01$; ns, not significant. (C) Magnitude and kinetics of serological response for four COVID-19 pneumonia patients. (D) Magnitude and kinetics of serological response for 10 COVID-19 patients with mild upper respiratory tract illness. RBD, receptor-binding domain; D, day after onset.

frequency of 1.3% ± 1.2% of peripheral B cells and were detected on day 19 ± 7 (6 to 33) after the onset of illness (Table S2). Among 20 samples (from 20 patients) tested, five were collected less than 2 weeks after the onset of illness, and three of them (60%) had no detectable memory B cell response; in contrast, only one from another 15 samples (7%) that were collected more

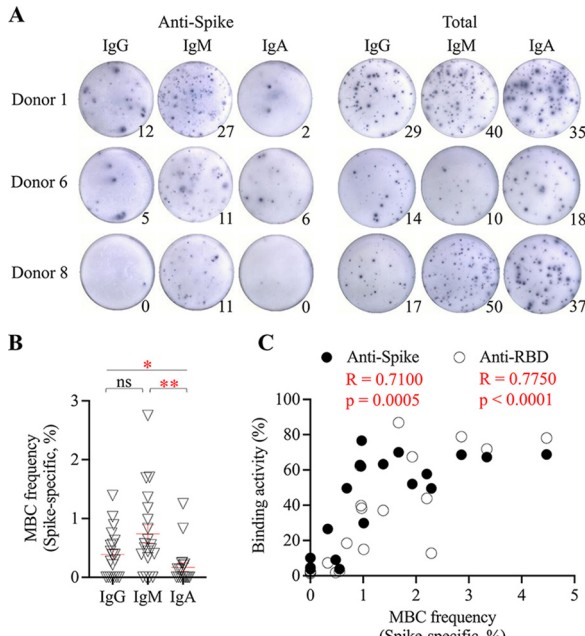

**FIG 2** Anti-spike memory B cell response to natural infection with SARS-CoV-2. (A) Enzyme-linked immunosorbent spot showing SARS-CoV-2 spike-specific IgG, IgM, and IgA memory B cell in the peripheral blood upon natural infection. Circulating total IgG, IgM, and IgA memory B cells were also measured. A total of 200,000 cultured cells were added into each of the anti-spike B cell wells, and 5,000 cultured cells were added into each of the total B cell wells. Donor 1 (day 27) was a 55-year-old woman with severe COVID-19 with pneumonia. Donor 6 (day 21) was a 43-year-old man with moderate COVID-19 and pneumonia. Donor 8 (day 18) was a 22-year-old woman with mild COVID-19 and a running nose. (B) Frequency of spike-specific IgG, IgM, and IgA memory B cells in the peripheral blood ($n = 20$). The frequency is defined as the percentage of spike-specific IgG (IgM or IgA) B cells in the total IgG (IgM or IgA) B cells. Each point represents memory B cell frequency for each patient, and the red line represents mean ± standard error of mean. One-way ANOVA with *post hoc* Tukey's test was used to compare the difference among groups. *, $P < 0.05$; **, $P < 0.01$. ns, not significant. (C) Relationship of spike-specific memory B cell frequency (IgG + IgM + IgA) and spike-binding and RBD-binding serological activity among COVID-19 patients. Linear regression was used to model the relationship between two variables. MBC, memory B cell.

than 2 weeks after the onset of illness failed to develop a memory B cell response (Fisher's exact test, $P = 0.02$) (Table S2). Although SARS-CoV-2 infection elicited a spike-specific B cell response, IgM memory B cells were predominantly induced, followed by IgG and IgA B cells (Fig. 2B).

Upon natural infection, the spike-specific memory B cell response was significantly correlated with the peak spike-binding and ACE2-blocking serological response, indicating a critical role of the B cell response in the development of antibody immunity upon SARS-CoV-2 infection (Fig. 2C).

**Relationship of antibody response and clinical severity.** Primary infection with SARS-CoV-2 can cause mild to severe clinical symptoms. In this study, fever duration did not correlate with the peak viral load in the respiratory sample (see Fig. S2A in the supplemental material). However, the fever duration was significantly correlated with the magnitude of spike-binding, RBD-binding antibody responses, and functional hemagglutination titers (Fig. S2B). Patients who experienced fever had a significantly stronger RBD-binding antibody response, hemagglutination titer, and spike-specific IgM B cell response than patients without fever (Fig. 3B). Patients who developed pneumonia showed significantly stronger anti-spike antibody and B cell responses than patients without pneumonia (Fig. 3A).

**Longevity of the antibody response.** The follow-up serum samples were collected from a subset of patients at least 7 months after infection (Table S1). Between enrollment and follow-up, no reinfection with SARS-CoV-2 was found among the patients. The spike-binding antibody response and functional hemagglutination titer waned at 11 ± 3 (7 to 15) months after infection (see Fig. S3 in the supplemental material). A substantial portion of the patients tested (75%, 6 of 8) had no detectable hemagglutination serological titer during follow-up, and an 8- to 16-fold reduction in titer was noted in the remaining patients.

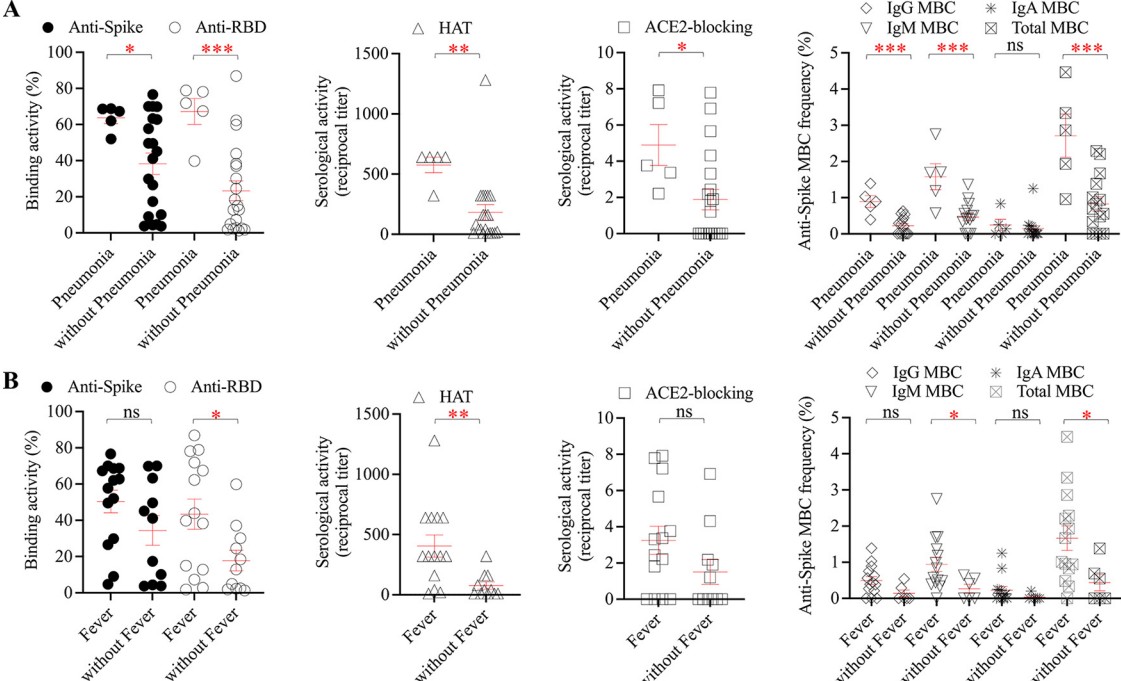

**FIG 3** Comparison of anti-spike antibody and B cell responses between patients with (*n* = 5) and without (*n* = 20) pneumonia (A) and patients with (*n* = 14) and without (*n* = 11) fever (B). Each point represents antibody or B cell response for each patient, and the red line represents mean ± standard error of mean. Unpaired two-tailed *t* test was used to compare the difference between two groups. *, $P < 0.05$; **, $P < 0.01$; ***, $P < 0.001$. ns, not significant. HAT, hemagglutination titer.

An average of 2.6 ± 1.0 (1.0 to 3.5)-fold reduction in the spike-binding antibody response was also detected, which was in accordance with the decline in the functional serological titer.

**Infection-induced anti-spike antibodies against emerging variants.** The neutralization activity of convalescent and follow-up serum samples was then tested against the Beta, Delta, and Omicron variant pseudoviruses. The results showed that infection-induced anti-spike antibodies had greatly reduced activities against Beta, Delta, and Omicron variants in both convalescent and follow-up serum samples (Fig. 4). The convalescent and follow-up serum samples showed 83 ± 82 (15 to 306)- and 165 ± 167 (12 to 456)-fold reductions in the neutralization activity against the Omicron variant, respectively, suggesting a substantial effect of multiple mutations in the Omicron spike on the antibody response upon natural infection.

There was a positive relationship between spike-binding antibody response and neutralizing activity in the convalescent-phase serum samples, but this trend did not reach statistical significance (see Fig. S4 in the supplemental material).

## DISCUSSION

SARS-CoV-2 belongs to the large family of coronaviruses, which includes viruses causing common cold (229E, NL63, OC43, and HKU1), Middle East respiratory syndrome coronavirus, and severe acute respiratory syndrome coronavirus. The process of antibody production is assumed to be similar to that against other seasonal coronaviruses. During SARS-CoV-2 infection, antibodies of the IgM class may be detected approximately 6 days after infection, and IgG may be detected after 8 days; concentrations of the antibodies may then decline over several months, allowing subsequent infection (17, 18). In this study, we clearly demonstrated that the anti-SARS-CoV-2 spike antibody level surged as early as the first week after symptom onset, continued to rise between the second and third weeks, and reached a plateau by 3 weeks during the convalescence period, in line with the findings of previous studies. Antibodies that recognize the RBD have been considered the most important component of immunity against SARS-CoV-2 in humans (6, 17–19). In this study, the RBD-binding antibody response, with its functional activities measured using hemagglutination

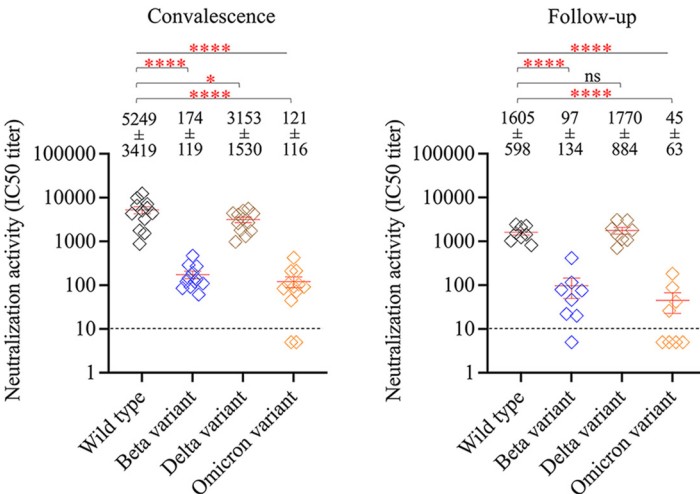

**FIG 4** Effect of SARS-CoV-2 variants on anti-spike antibody response to natural infection. Neutralization activities of convalescent-phase serum samples ($n = 12$, left) and follow-up serum samples ($n = 8$, right) against wild type, Beta variant, Delta variant, and Omicron variant pseudoviruses. All serum samples were tested with a starting dilution of 1:10 (cutoff, dashed line), and those that failed to neutralize virus at the starting dilution were recorded as half-maximal inhibitory concentration ($IC_{50}$) titer 5. Each point represents $IC_{50}$ titer for each sample, and the red line represents mean ± standard error of mean. One-way ANOVA with *post hoc* Tukey's test was used to compare the difference among groups. *, $P < 0.05$; ****, $P < 0.0001$; ns, not significant.

and the ACE2-blocking assay, was detectable simultaneously as spike-binding antibodies were elicited. This implied that the detected anti-RBD antibodies following natural infection with wild-type SARS-CoV-2 might be neutralizing. This was further corroborated in this study by serological neutralizing activity against viruses bearing the wild-type and variant spike proteins.

Memory B cells circulate throughout the body in a quiescent state and are critical in an accelerated secondary immune response upon reexposure to the pathogen (20, 21). Therefore, development of the memory B cell response to the spike protein constitutes a major part of humoral immunity against SARS-CoV-2. A previous study reported that the size of RBD-specific memory B cells may remain stable nearly 6 months after natural infection with SARS-CoV-2 and that these memory cells could further display clonal turnover, and the derived antibodies exhibit resistance to RBD mutations (22, 23). Here, our results demonstrated that the spike-specific memory B cell response was detectable 3 weeks after the onset of illness and that the memory B cell response significantly correlated with peak spike-binding and ACE2-blocking serological response, indicating a critical role of B cell response. Although, the role of T cell-mediated immune response was not explored in this study, the generation of virus-specific memory B cells after SARS-CoV-2 infection could be dependent on the presence of CD4[+] T helper cells (10, 24).

Primary infection with SARS-CoV-2 causes mild to severe clinical symptoms. In previous studies, patients with severe COVID-19 developed a stronger antibody response than those with mild illness (18, 25). Consistently, we observed that patients who experienced fever had a significantly stronger RBD-binding antibody response, hemagglutination titer, and spike-specific IgM B cell response than those without fever. Patients who developed pneumonia showed significantly stronger anti-spike antibody and B cell responses than patients without pneumonia. While these results are compatible with the findings of previous studies (18, 25), the mechanisms underlying the relationship between the magnitude of anti-spike antibody or B cell response and the disease severity remains largely unclear. Current evidence suggests that the elicitation of anti-spike antibodies would contribute to protective immunity against subsequent infection rather than an increased risk of antibody-dependent enhancement (26), but further studies on immunity waning and the possible existence of antibody-dependent enhancement are required in the future.

The magnitude of the spike-binding antibody level was more than 2-fold lower after an average of 11 months than that in the convalescent stage of infection. Thus, passive

reimmunization through vaccination could be beneficial to boost anti-spike antibody level among individuals with prior infection.

Emerging SARS-CoV-2 variants could spread quickly and become the dominant strains during outbreaks (27). For example, the Beta (B.1.351) and Delta (B.1.617.2) variants were designated as variants of concern (VOCs) in 2021 (28). At the end of 2021, the Omicron variant (B.1.1.529) emerged and rapidly cocirculated with the Delta variant (29, 30). Recently, the Omicron variant has become prevalent worldwide (31). These variants have several mutations in the spike protein, with the Omicron (B.1.1.529) clade displaying over 30 changes, 15 of which are located in the RBD (32). Further evidence indicates that spike mutations in emerging variants may enhance transmissibility and contribute to the escape of viruses from humoral immunity (33, 34). Recent research has demonstrated that convalescent-phase serum samples do not provide cross-protection against new Omicron variants (35, 36). Congruously, our study demonstrated decreased neutralization activity, with more than 80- and 160-fold reduction against the Omicron variant in convalescent and follow-up serum samples, respectively. It has been shown that a booster immunization may elicit a prominent neutralization titer against the variant, which may reduce the risk of breakthrough infection (36, 37).

Some limitations may exist in the study. First, this was a single-center observational study, the number of patients was small, and the time points of blood sampling varied among patients. In the study, a total of 48 convalescent-phase serum samples were collected from 25 patients with COVID-19 during hospitalization. While 11 patients provided samples at a single time point, 14 patients provided multiple samples during hospitalization (see Table S1 in the supplemental material). Magnitude and kinetics of serological response for these 14 patients have been shown in Fig. 2. After discharge, follow-up was arranged, but 17 patients were lost to follow-up or declined to provide the blood sample. Finally, eight follow-up serum samples were collected from eight patients (see Tables S1 and S2 in the supplemental material). Thus, there is limited information about anti-spike antibody response during the follow-up period. Second, anti-spike antibody response was undetectable in four patients. Most of them did not have risk factors, i.e., immunocompromised status, obesity, and age over 65 years, that may affect immunological responses (38, 39). However, all four patients had a mild illness, and one of them provided samples within the first week of illness. Third, a flow cytometry-based binding assay was used on the basis of testing a single dilution of serum to quantify the level of spike- and RBD-binding antibodies elicited by acute SARS-CoV-2 infection in the study. A similar assay and sample dilution have been used to test anti-spike antibody response in a previous study (19, 40–42). We did not perform serum titrations to acquire the end-points in our binding assay, and our data were obtained from the percentage of antibody-bound spike (or RBD)-expressing cells at a single serum dilution (1:100). Although the focus of the present study was to examine the kinetics and magnitude of spike-specific antibody response, and not to define antibody titer as a correlating factor of immune protection, our findings on spike- and RBD-binding activities in the serum need to be interpreted with caution. Fourth, a longitudinal follow-up of memory B cell responses was not performed, as only serum samples were available then. Previous research has revealed that the size of antigen-specific memory B cell repertoire may show a trajectory pattern, and a long-term observation might be required to assess the dynamics of anti-spike memory B cell populations in the near future (17, 18, 22, 23, 25). Finally, we did not identify the viral strains of these 25 patients; thus, whether any of them may be infected with VOCs is not known. Wuhan strain is the reasonable one based on the epidemiology studies between February and September 2020 in Taiwan and worldwide (43, 44).

In conclusion, this study confirmed that acute SARS-CoV-2 infection elicits a rapid and robust spike-binding and ACE2-blocking antibody response, which wanes approximately 11 months after infection. Serological responses correlate with the frequency of spike-specific memory B cell responses to natural infections. Patients with fever and pneumonia develop significantly higher spike-binding, ACE2-blocking, and memory B cell responses. However, spike-specific antibody responses are greatly affected by spike mutations in emerging variants, especially the Beta and Omicron variants. These results warrant continued surveillance

of the spike-specific antibody response to natural infection and suggest the maintenance of functional anti-spike antibodies through vaccination.

## MATERIALS AND METHODS

**Ethics statement.** The study protocol and informed consent form were approved by the ethics committee of Chang Gung Medical Foundation and Taoyuan General Hospital, Ministry of Health and Welfare. Written informed consent was obtained from each participant before inclusion in the study. The study and all associated methods were performed in accordance with the approved protocol, the principles of the Declaration of Helsinki, and Good Clinical Practice guidelines.

**Patient enrollment and sample collection.** Patients who were diagnosed with acute SARS-CoV-2 infection (COVID-19) using real-time reverse transcriptase-PCR (rRT-PCR) analysis of oropharyngeal swab samples between January and September 2020 were enrolled. Patients were hospitalized in a negative-pressure isolation room according to the regulations of the Taiwan Centers for Disease Control. Convalescent blood samples were collected from the enrolled patients during hospitalization. Patients were discharged from hospital when they had three consecutive negative rRT-PCR test results and resolved symptoms. Thereafter, the patients underwent follow-up care in the outpatient clinics and provided follow-up blood samples. The serum samples were stored at $-20°C$ before testing.

**Collection of respiratory samples and measurement of viral load.** Briefly, 300 $\mu$L of oropharyngeal swab samples were subjected to nucleic acid extraction using the LabTurbo kit (Taigen Bioscience, Taipei, Taiwan) on a LabTurbo 48 compact system (Taigen Bioscience, Taipei, Taiwan) (45). The isolated nucleic acids were eluted in 50 $\mu$L of elution buffer. RT-PCR was performed using the LightCycler Multiplex RNA virus master kit (Roche Diagnostics, Mannheim, Germany) on a Cobas z480 analyzer (Roche Diagnostics, Mannheim, Germany), targeting envelope (E), nucleocapsid (N), and RNA-dependent RNA polymerase (RdRp) genes (46) along with an internal control. The specific primers and probes were from the ModularDx kit (Tib Molbiol, Berlin, Germany). Negative samples had cycle threshold ($C_T$) values higher than 37 in the reactions of both E and RdRp genes.

**Flow cytometry.** Flow cytometry-based binding assay was used to examine serological anti-spike and anti-RBD-binding activity. To perform a flow cytometry-based binding assay, transduced MDCK-SIAT1 cells expressing Wuhan-Hu-1 RBD (MDCK-RBD) or spike (MDCK-spike) were prepared (19). In brief, each serum sample was tested in duplicate at a dilution of 1:100 in phosphate-buffered saline (PBS)-3% bovine serum albumin (BSA), and 100 $\mu$L of sample was incubated with $3 \times 10^5$ MDCK-RBD or MDCK-spike cells at 4°C for 30 min. After washing with PBS, the cells were incubated with fluorescein isothiocyanate (FITC)-conjugated anti-human IgG secondary antibody at 4°C for 30 min. After washing with PBS, the cells were analyzed using a BD FACSCanto II flow cytometer. The cells in forward scatter (FSC)/side scatter (SSC) gate were selected, and FITC-positive cells were further identified and gated. At least 5,000 events were acquired for the analysis. For each experiment, the cells incubated with only PBS-3%BSA were included as a negative control. Other controls included cells incubated with anti-influenza H3 human IgG antibody BS-1A (1 $\mu$g/mL) (produced in-house) and with anti-SARS-CoV-2 RBD human IgG antibody EY-6A (1 $\mu$g/mL) or FI-3A (1 $\mu$g/mL) (6). Serological anti-spike binding activity was determined by the percentage of antibody-bound (FITC-positive) spike-expressing cells in serum minus the percentage of antibody-bound (FITC-positive) spike-expressing cells in PBS control. Serological anti-RBD-binding activity was determined by the percentage of antibody-bound (FITC-positive) RBD-expressing cells in serum minus the percentage of antibody-bound (FITC-positive) RBD-expressing cells in PBS control.

**Hemagglutination assay.** The RBD-binding activity of serum samples was analyzed using a hemagglutination assay in 96-well U-bottom plates. Serial dilutions of serum samples or monoclonal antibodies (200 $\mu$g/mL) beginning at 1:20 dilution in PBS were mixed with VHH(IH4)-RBD (47) at room temperature. Thereafter, human type O erythrocytes in PBS at 1:40 dilution were added to the mixture in each well and incubated at room temperature for another 60 min. The plate was tilted at 45° for at least 30 s and then examined. Anti-RBD antibodies in the serum react with VHH(IH4)-RBD and facilitate cross-linking of erythrocytes and agglutination occurs. The endpoint was defined as the final dilution producing visible agglutination. Controls included anti-SARS-CoV-2 RBD human IgG antibody EY-6A or FI-3A (19, 47). A hemagglutination titer of 1:20 or more was considered to be positive. Samples with a reciprocal titer less than 20 were assigned a value of 10.

**ACE2-blocking assay.** A flow cytometry-based assay was used to analyze the RBD-ACE2-blocking activity of the serum samples (19). Two-fold serial dilutions of original serum samples in PBS were mixed with biotinylated RBD at room temperature. The mixture was then incubated with human ACE2-expressing MDCK-SIAT1 (MDCK-ACE2) cells at 4°C for 30 min. After washing, the ExtrAvidin-R-phycoerythrin protein was incubated with the cells at 4°C for 30 min. After washing, the cells were analyzed using a BD FACSCanto II flow cytometer. At least 5,000 events of RBD-bound (phycoerythrin-positive) cells were acquired for the analysis. The PBS-biotinylated RBD mixture was used to obtain maximum signal, and PBS only was used to determine background. The original, undiluted serum that failed to inhibit ACE2-RBD interaction was scored as negative. The ACE2-blocking titer is expressed as the reciprocal of the serum dilution presenting 50% inhibition of signal compared to the maximum signal.

**Pseudovirus neutralization assay.** HEK293T cells stably expressing human ACE2 were seeded in a 96-well plate and incubated overnight. Serial dilutions of heat-inactivated serum samples were prepared and mixed with pretitrated pseudotyped lentiviruses expressing the wild-type Wuhan-1, Beta variant, Delta variant, or Omicron variant spike proteins at 37°C for 1 h. They were then inoculated into the preseeded cells at 37°C, and incubated for another 16 h. The culture medium was then replaced with fresh Dulbecco's modified Eagle medium (DMEM) supplemented with 1% fetal bovine serum and 100 U/mL penicillin/streptomycin. After incubating for another 48 h, luciferase activity was measured using the Bright-Glo luciferase assay system (Promega, United States). A virus control was included for each assay. The inhibitory activity at each serum dilution was

determined according to the relative light unit value as follows: [(relative light unit$_{Control}$ − relative light unit$_{serum}$)/ relative light unit$_{Control}$] × 100.

**Memory B cell assay.** Peripheral blood mononuclear cells (PBMCs) were separated using Ficoll lymphocyte separation medium. Resuspended PBMCs were cultured in complete medium containing pokeweed mitogen (PWM), *Staphylococcus aureus* Cowan I, and CpG at 37°C for 5 days. Cultured PBMCs were collected, washed, and resuspended for the ELISpot assay. The ELISpot assay was used to detect spike-binding IgG-, IgM-, and IgA-secreting cells. Briefly, a 96-well Millipore plate was coated with SARS-CoV-2 spike or anti-human IgG at 4°C overnight. After washing, the plates were blocked with 2% dry skim milk at 37°C for 2 h. After washing, resuspended cultured cells were added to the wells and incubated at 37°C for 16 h. After washing, the wells were incubated with anti-human IgG, IgM, or IgA secondary antibodies conjugated with alkaline phosphatase at room temperature for 2 h. After washing, the wells were developed using an alkaline phosphatase substrate kit at room temperature for 2 to 5 min. The spot-forming cells were counted using an automated ELISpot plate reader (48).

**Statistical analyses.** Statistical analyses were performed using GraphPad Prism or Excel. The unpaired *t* test was used to determine the differences between independent groups. A one-way analysis of variance was used to determine the differences among three or more independent groups, and Tukey's test was used for *post hoc* analysis. The chi-squared test was used to analyze the relationships between categorical variables. Linear regression was used to evaluate the correlation between variables. Statistical significance was set at a *P* of $<0.05$.

## SUPPLEMENTAL MATERIAL

Supplemental material is available online only.
**SUPPLEMENTAL FILE 1**, PDF file, 0.2 MB.

## ACKNOWLEDGMENTS

We acknowledge Alain Townsend, Tiong Kit Tan, Pramila Rijal, and Lisa Schimanski for the support of VHH(IH4)-RBD. We also thank the patients and the care team of the COVID-19-designated wards at Taoyuan General Hospital, Ministry of Health and Welfare.

We report no potential conflict of interest.

This work was mainly supported by a grant (BMRPE22) from the Chang Gung Memorial Hospital and a grant (MOST 110-2628-B-182-013) from Ministry of Science and Technology of Taiwan to Kuan-Ying A. Huang.

Conceptualization, Cheng-Pin Chen, Kuan-Ying A. Huang, and Shu-Hsing Cheng; data curation, Cheng-Pin Chen, Kuan-Ying A. Huang, Yi-Chun Lin, Chien-Yu Cheng, and Shu-Hsing Cheng; formal analysis, Kuan-Ying A. Huang, Shin-Ru Shih, and Shu-Hsing Cheng; funding acquisition, Kuan-Ying A. Huang; investigation, Cheng-Pin Chen, Kuan-Ying A. Huang, and Shu-Hsing Cheng; methodology, Kuan-Ying A. Huang and Shin-Ru Shih; project administration, Kuan-Ying A. Huang and Shu-Hsing Cheng; resources, Kuan-Ying A. Huang and Shu-Hsing Cheng; software, Kuan-Ying A. Huang; supervision, Yhu-Chering Huang and Tzou-Yien Lin; validation, Cheng-Pin Chen, Kuan-Ying A. Huang, and Shu-Hsing Cheng; visualization, Cheng-Pin Chen and Kuan-Ying A. Huang; writing – original draft, Cheng-Pin Chen, Kuan-Ying A. Huang, and Shu-Hsing Cheng; writing – review and editing, Cheng-Pin Chen, Kuan-Ying A. Huang, Shin-Ru Shih, Yi-Chun Lin, Chien-Yu Cheng, Yhu-Chering Huang, Tzou-Yien Lin, and Shu-Hsing Cheng.

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
