## [Reviewer comments · Microbiology Spectrum]

Microbiology Spectrum

Anti-spike antibody response to natural infection with SARS-CoV-2 and its activity against emerging variants

Cheng-Pin Chen, Kuan-Ying Huang, Shin-Ru Shih, Yi-Chun Lin, Chien-Yu Cheng, Yhu-Chering Huang, Tzou-Yien Lin, and Shu-Hsing Cheng

Corresponding Author(s): Shu-Hsing Cheng, Taoyuan General Hospital, Ministry of Health Welfare, Taoyuan, Taiwan; School of Public Health, College of Public Health and Nutrition, Taipei Medical University, Taipei, Taiwan

Review Timeline:

Submission Date:	February 26, 2022
Editorial Decision:	March 23, 2022
Revision Received:	April 25, 2022
Editorial Decision:	May 10, 2022
Revision Received:	May 18, 2022
Accepted:	May 24, 2022

Editor: Takamasa Ueno

Reviewer(s): The reviewers have opted to remain anonymous.

Transaction Report:

DOI: <https://doi.org/10.1128/spectrum.00743-22>

March 23, 2022

Dr. Shu-Hsing Cheng

Taoyuan General Hospital, Ministry of Health Welfare, Taoyuan, Taiwan; School of Public Health, College of Public Health and Nutrition, Taipei Medical University, Taipei, Taiwan
Department of Infectious Diseases
Taipei
Taiwan

Re: Spectrum00743-22 (Anti-spike antibody response to natural infection with SARS-CoV-2 and its activity against emerging variants)

Dear Dr. Shu-Hsing Cheng:

Link Not Available

Sincerely,

Takamasa Ueno

Journals Department
Reviewer comments:

Reviewer #1 (Comments for the Author):

The authors investigated the serological activities, such as anti-spike antibody response, hemagglutination inhibition activity, ACE2 blocking activity, and pseudovirus neutralization activity, against SARS-CoV-2 in sera from patients with COVID-19. They also investigated the relationship between serological activities and spike-specific memory B cell response. Furthermore, they examined the effect of the emerging variants, including the Omicron variant, on pseudovirus neutralization activity in convalescence and follow-up sera.

They revealed that the serological responses were correlated with the frequency of spike-specific memory B cell responses.

Patients with fever and pneumonia showed higher spike-binding, ACE2-blocking antibodies, and memory B cell responses. Furthermore, they showed that the pseudovirus neutralization activity was substantially affected by the emerging variants, especially the Beta and Omicron variants.

This work has been performed well and described in a clear manner; however, some points need to be addressed.

1. X-axis labeling in figures: "Serological activity (%)" is not clear. Is it the percentage of spike- or RBD-positive cells obtained by flow cytometry analysis? Please provide what this labeling indicates in the Methods section and figure legends.
2. Measurement of antibody titer is one of the easiest ways to assess serological activity in serum samples. Why did the authors not measure antibody titers against spike or RBD? Please discuss about it or include the data of antibody titers.
3. Figure 4 and Figure S3: Although 25 patients were enrolled in this study, only eight convalescent and 12 follow-up sera were examined for neutralization activity. Also, when were the convalescent and follow-up sera collected? Please explain why only a subset of patients could be analyzed and provide more detailed information about these samples in the methods section.
4. Table S1: Please include the abbreviation for URI in the legend.

Reviewer #2 (Comments for the Author):

Chen et al. analyzed antibody response in COVID-19 patients. Antibodies to bind Spike or RBD, antibodies to block RBD-ACE2 binding and Spike-specific memory B cells were analyzed. Antibody response was high in patients with fever or pneumonia. Neutralizing activity of sera samples were high against wildtype and delta variant, but low against beta and omicron. Major concern is low number of samples analyzed. Total of 25 patients are listed in Table 1, but multiple samples were not obtained from most of patients. Similar studies using large number of patients were published as authors mentioned, and no new finding was shown in this manuscript.

Correlation between anti-spike Ab response and pneumonia or fever is shown, but no further analysis was performed to search the cause of this correlation. Memory B cell data are interesting, but the detail is not shown in this manuscript. Memory B cell data at follow-up will be useful to estimate maintenance of antibody response to SARS-CoV-2 infection. Although neutralization of variants are shown in the last section, relation of these data to other results are unclear.

Other concerns

1) Importance section is largely the repetition of abstract.

2) line 176; "most patients were from other countries/regions."
Is this mean that patients were not Taiwanese?

3) Figure 1

It is unclear how serological activity (%) is measured. HAT, haemagglutination-inhibition titer, is not adequate explanation, because this is inhibition of RBD-antibody binding.

4)line 190; "The spike-binding antibody response was detected as early as the first week after onset, continued to rise between the second and third weeks, and reached a plateau at three weeks after onset during the convalescence period (Figure 1B, 1C)."

Only 5 samples for the first week were shown in Figure 1B and 1C. Sample number is too few to conclude the detection at the first week. The rise and plateau are also hard to speculate from these data.

5) line 212; "The memory B cell response was detectable within two weeks ($n = 5$, $0.6 \pm 1.0\%$, day 6-13) and increased three weeks after the onset of illness ($n = 15$, $1.5 \pm 1.2\%$, day 14-33), but the difference was not statistically significant ($p = 0.12$, unpaired two-tailed t-test). Three patients (60%) had no detectable memory B cell response in the first two weeks after onset; in contrast, one patient (7%) failed to develop a memory B cell response even at three weeks after onset. Although SARS-CoV-2 infection elicited a spike-specific B cell response, IgM memory B cells were predominantly induced, followed by IgG and IgA B cells (Figure 2B)."

Line 212 should be the following.

"Memory B cell response was detectable within two weeks in two of five patients."

Percentages are unclear. Three patients (60%) and one patient (7%) are in the same sentence.

Data for these sentences are not shown. Time points and patient information are not available in figure 2. X axis of figure 2C should be "MBC frequency (%)"

6) Figure 4

Information of virus strain in patients should be provided.

Staff Comments:

Preparing Revision Guidelines

Please return the manuscript within 60 days; if you cannot complete the modification within this time period, please contact me. If you do not wish to modify the manuscript and prefer to submit it to another journal, please notify me of your decision immediately so that the manuscript may be formally withdrawn from consideration by Microbiology Spectrum.

The authors investigated the serological activities, such as anti-spike antibody response, hemagglutination inhibition activity, ACE2 blocking activity, and pseudovirus neutralization activity, against SARS-CoV-2 in sera from patients with COVID-19. They also investigated the relationship between serological activities and spike-specific memory B cell response. Furthermore, they examined the effect of the emerging variants, including the Omicron variant, on pseudovirus neutralization activity in convalescence and follow-up sera.

They revealed that the serological responses were correlated with the frequency of spike-specific memory B cell responses. Patients with fever and pneumonia showed higher spike-binding, ACE2-blocking antibodies, and memory B cell responses. Furthermore, they showed that the pseudovirus neutralization activity was substantially affected by the emerging variants, especially the Beta and Omicron variants.

This work has been performed well and described in a clear manner; however, some points need to be addressed.

1. X-axis labeling in figures: “Serological activity (%)” is not clear. Is it the percentage of spike- or RBD-positive cells obtained by flow cytometry analysis? Please provide what this labeling indicates in the Methods section and figure legends.

2. Measurement of antibody titer is one of the easiest ways to assess serological activity in serum samples. Why did the authors not measure antibody titers against spike or RBD? Please discuss about it or include the data of antibody titers if available.

3. Figure 4 and Figure S3: Although 25 patients were enrolled in this study, only eight convalescent and 12 follow-up sera were examined for neutralization activity. Also, when were the convalescent and follow-up sera collected? Please explain why only a subset of patients could be analyzed and provide more detailed information about these samples in the methods section.

4. Table S 1: Please include the abbreviation for URI in the legend.

Dear Editor and Reviewers:

We thank you for your valuable comments and suggestions. We have revised the manuscript accordingly; our responses to the comments are provided below.

Reviewer #1 (Comments to the Authors)

The authors investigated the serological activities, such as anti-spike antibody response, hemagglutination inhibition activity, ACE2 blocking activity, and pseudovirus neutralization activity, against SARS-CoV-2 in sera from patients with COVID-19. They also investigated the relationship between serological activities and spike-specific memory B cell response. Furthermore, they examined the effect of the emerging variants, including the Omicron variant, on pseudovirus neutralization activity in convalescence and follow-up sera. They revealed that the serological responses were correlated with the frequency of spike-specific memory B cell responses. Patients with fever and pneumonia showed higher spike-binding, ACE2-blocking antibodies, and memory B cell responses. Furthermore, they showed that the pseudovirus neutralization activity was substantially affected by the emerging variants, especially the Beta and Omicron variants. This work has been performed well and described in a clear manner; however, some points need to be addressed.

1. X-axis labeling in figures: "Serological activity (%)" is not clear. Is it the percentage of spike- or RBD-positive cells obtained by flow cytometry analysis? Please provide what this labeling indicates in the Methods section and figure legends.

- *Serological activity was assessed by measuring anti-spike and anti-RBD binding activities in the serum samples. Anti-spike binding activity was determined by the percentage of antibody-bound spike-expressing cells in serum minus the percentage of antibody-bound spike-expressing cells in PBS control. Anti-RBD binding activity was determined by the percentage of antibody-bound RBD-expressing cells in serum minus the percentage of antibody-bound RBD-expressing cells in PBS control. We have further clarified this in the methods section and figure legends in the revised manuscript (page 7, and fig.1).*

2. Measurement of antibody titer is one of the easiest ways to assess serological activity in serum samples. Why did the authors not measure antibody titers against spike or RBD? Please discuss about it or include the data of antibody titers.

- *A flow cytometry-based binding assay was used on the basis of testing a single dilution of serum to quantify spike- and RBD-binding antibodies elicited by acute SARS-CoV-2 infection in the study. A similar assay and sample dilution have been used to test anti-spike antibody response in previous studies [refs 19,44-46]. We did not perform serum titrations to acquire the end-points in our binding assay, and our data were obtained from the percentage of antibody-bound spike (or RBD)-expressing cells at a single serum dilution (1:100). The focus of the present study was*

to examine the kinetics and magnitude of spike-specific antibody response, and not to define antibody titer as a correlate of immune protection. We have further discussed this limitation of the study and mentioned that our findings on spike- and RBD-binding activities in the serum need to be interpreted with caution, in the revised manuscript (page 17). We have provided the details of flow cytometry-based binding assay in the methods section in the revised manuscript (page 7).

3. Figure 4 and Figure S3: Although 25 patients were enrolled in this study, only eight convalescent and 12 follow-up sera were examined for neutralization activity. Also, when were the convalescent and follow-up sera collected? Please explain why only a subset of patients could be analyzed and provide more detailed information about these samples in the methods section.

- In the study, 48 convalescent serum samples were collected from 25 patients with COVID-19 during hospitalization. For each patient, at least one serum sample was collected and 14 patients provided additional convalescent sera during hospitalization (Table S1). The patients were discharged from hospital when they had three consecutive negative rRT-PCR test results and resolved symptoms. Thereafter, patients underwent follow-up care in the outpatient clinics and provided follow-up blood samples. During the follow-up period in the outpatient clinics, 17 patients were lost to follow-up or declined to provide blood sample. Thus, eight follow-up serum samples were collected from eight patients (Table S1). We have provided more details about the collection of convalescent and follow-up sera in the revised manuscript (pages 6, 11, 17; Table S1 and S2).*
- Neutralization test was arranged in the late stage of study and at that time the volume of several convalescent sera was not sufficient for the neutralization test. Twelve convalescent serum samples were available and tested using the neutralization assay. Besides, all eight follow-up serum samples were tested using the neutralization assay. We have provided more details about the collection of convalescent and follow-up sera in the revised manuscript (pages 6, 11, 17; Table S1 and S2).*

4. Table S1: Please include the abbreviation for URI in the legend.

- We have defined the abbreviation URI in the footnote of Table S1.*

Reviewer #2 (Comments for the Author):

Chen et al. analyzed antibody response in COVID-19 patients. Antibodies to bind Spike or RBD, antibodies to block RBD-ACE2 binding and Spike-specific memory B cells were analyzed. Antibody response was high in patients with fever or pneumonia. Neutralizing activity of sera samples were high against wildtype and delta variant, but low against beta and omicron.

Major concern is low number of samples analyzed. Total of 25 patients are listed in Table 1, but multiple samples were not obtained from most of patients. Similar studies using large number of patients were published as authors mentioned, and no new finding was shown in this manuscript.

Correlation between anti-spike Ab response and pneumonia or fever is shown, but no further analysis was performed to search the cause of this correlation. Memory B cell data are interesting, but the detail is not shown in this manuscript. Memory B cell data at follow-up will be useful to estimate maintenance of antibody response to SARS-CoV-2 infection. Although neutralization of variants are shown in the last section, relation of these data to other results are unclear.

- *The limitation of a small number of patients analyzed in the study has been further discussed in the revised manuscript (page 17). Other similar studies have been cited (refs. 23, 29).*
- *While the results of correlation between anti-spike antibody response and pneumonia or fever are compatible with the findings of previous studies (refs. 23, 29), the underlying mechanisms remain largely unclear. Further studies are required to elucidate the role of inflammatory cells or cytokine production in the development of spike-specific antibody response among patients with pneumonia or prolonged fever in the near future, but this is beyond the scope of the current study. We have further discussed this finding in the revised manuscript (pages 15,16).*
- *We agree that the information about spike-specific memory B cell response during the follow-up period in patients with COVID-19 would be helpful to delineate the maintenance of B cell response to SARS-CoV-2. However, only serum samples were available in the follow-up period and the cellular response information was lacking in the present study. We have further discussed this study limitation in the revised manuscript (page 17).*
- *We further analyzed the relation between neutralizing activity and other antibody response profiles and provided related data in the revised manuscript (page 14, Fig. S4).*

Other concerns

1) Importance section is largely the repetition of abstract.

- *We have revised the Importance section in the manuscript (pages 3, 4).*

2) line 176; "most patients were from other countries/regions."
Is this mean that patients were not Taiwanese?

- *Among the 25 enrolled patients, 23 were Taiwanese adults and 2 were foreign adults. All were identified as imported cases of COVID-19 by Taiwan Centers for Disease Control. We have clarified the origin of these patients in the revised manuscript (page 10, line 209).*

3) Figure 1

It is unclear how serological activity (%) is measured. HAT, haemagglutination-inhibition titer, is not adequate explanation, because this is inhibition of RBD-antibody binding.

- *Serological activity was assessed by measuring anti-spike and anti-RBD binding activities in the serum samples. Anti-spike-binding activity was determined by the percentage of antibody-bound spike-expressing cells in serum minus the percentage of antibody-bound spike-expressing cells in PBS control. Anti-RBD binding activity was determined by the percentage of antibody-bound RBD-expressing cells in serum minus the percentage of antibody-bound RBD-expressing cells in PBS control. We have further clarified this in the methods section and figure legends in the revised manuscript (page 7, and fig.1).*
- *We apologize for using a wrong term for the HAT assay. HAT is a hemagglutination assay for the presence of antibodies to the RBD of the SARS-CoV-2 spike protein in serum samples, using type O red cells as indicators. It has been shown that HAT is a surrogate marker for neutralizing antibodies (<https://doi.org/10.1038/s43856-022-00091-x>). We have provided more details of the hemagglutination test in the methods section and corrected the related description in the revised manuscript (page 8).*

4)line 190; "The spike-binding antibody response was detected as early as the first week after onset, continued to rise between the second and third weeks, and reached a plateau at three weeks after onset during the convalescence period (Figure 1B, 1C)." Only 5 samples for the first week were shown in Figure 1B and 1C. Sample number is too few to conclude the detection at the first week. The rise and plateau are also hard to speculate from these data.

- *A total of 48 convalescent serum samples were collected from 25 adult patients with COVID-19 from days 5 to 39 after the onset of symptoms (Table S1, Table S2). We further analyzed this set of post-infection sera (5, 11, 12, and 20 serum samples collected in the 1st, 2nd, and 3rd weeks and after the 4th week of illness) and showed spike- and RBD-binding activities in the serum at different timepoints after onset in the revised manuscript (page 11 line 229, Fig. 1B).*
- *The spike-binding antibody response was detected as early as the first week after illness onset (binding activity $17\% \pm 8\%$, $n = 5$), continued to rise between the second (binding activity $39\% \pm 8\%$, $n = 11$) and third weeks (binding activity $51\% \pm 7\%$, $n = 12$)*

(ANOVA with post hoc, $p < 0.05$), and plateaued 4 weeks after onset (binding activity $52\% \pm 4\%$, $n = 20$) (ANOVA with post hoc, $p < 0.01$) during the convalescence period (Fig. 1B, C and D). In most cases, the RBD-binding antibody response was detectable at the same time point, whereas the spike-binding antibodies were elicited (Fig. 1; Table S2).

5) line 212; "The memory B cell response was detectable within two weeks ($n = 5$, 0.6 {plus minus} 1.0% , day 6-13) and increased three weeks after the onset of illness ($n = 15$, 1.5 {plus minus} 1.2% , day 14-33), but the difference was not statistically significant ($p = 0.12$, unpaired two-tailed t-test). Three patients (60%) had no detectable memory B cell response in the first two weeks after onset; in contrast, one patient (7%) failed to develop a memory B cell response even at three weeks after onset. Although SARS-CoV-2 infection elicited a spike-specific B cell response, IgM memory B cells were predominantly induced, followed by IgG and IgA B cells (Figure 2B)."

Line 212 should be the following.

"Memory B cell response was detectable within two weeks in two of five patients."

Percentages are unclear. Three patients (60%) and one patient (7%) are in the same sentence. Data for these sentences are not shown. Time points and patient information are not available in figure 2. X axis of figure 2C should be "MBC frequency (%)"

- We have provided more details (i.e., sample and patient information) of memory B cell response in Table S2. We have also revised the description of memory B cell response in the manuscript. We have revised the X axis label in Fig. 2C.*
- Among 20 samples (from 20 patients) tested, five were collected less than 2 weeks after the onset of illness and three of them (60%) had no detectable memory B cell response; In contrast, only one from other 15 samples (7%) that were collected more than 2 weeks after the onset of illness failed to develop a memory B cell response (Fisher's exact test, $p = 0.02$) (Page 12 line 250, Table S2).*
- Time points and the patient information of donor 1, 6 and 8 were added on the legends in fig. 2 as follows: Donor 1 (day 27): 55-y-o woman with severe COVID-19 with pneumonia. Donor 6 (day 21): 43-y-o man with moderate COVID-19 and pneumonia. Donor 8 (day 18): 22-y-o woman with mild COVID-19 and running nose.*

6) Figure 4

Information of virus strain in patients should be provided.

- We revised the study limitation and stated that we did not identify the viral strains of these 25 patients, thus if any of them may be infected with VOCs is not known. Wuhan strain is the reasonable one based on the epidemiology studies between Feb and Sep 2020 in Taiwan and worldwide [47,48] (page 18).*

May 9, 2022

Dr. Shu-Hsing Cheng

Taoyuan General Hospital, Ministry of Health Welfare, Taoyuan, Taiwan; School of Public Health, College of Public Health and Nutrition, Taipei Medical University, Taipei, Taiwan

Department of Infectious Diseases

1492 Jhongshan Rd

Taoyuan

Taiwan

Re: Spectrum00743-22R1 (Anti-spike antibody response to natural infection with SARS-CoV-2 and its activity against emerging variants)

Dear Dr. Shu-Hsing Cheng:

Please address the concern raised by Reviewer #2. I would suggest to modify or rephrase the term 'the original concentration'.

Thank you for submitting your manuscript to Microbiology Spectrum. As you will see your paper is very close to acceptance. Please modify the manuscript along the lines I have recommended. As these revisions are quite minor, I expect that you should be able to turn in the revised paper in less than 30 days, if not sooner. If your manuscript was reviewed, you will find the reviewers' comments below.

When submitting the revised version of your paper, please provide (1) point-by-point responses to the issues I raised in your cover letter, and (2) a PDF file that indicates the changes from the original submission (by highlighting or underlining the changes) as file type "Marked Up Manuscript - For Review Only". Please use this link to submit your revised manuscript. Detailed instructions on submitting your revised paper are below.

Link Not Available

Sincerely,

Takamasa Ueno

Reviewer comments:

Reviewer #1 (Comments for the Author):

All the comments I made on the first version have all been taken into consideration.

Reviewer #2 (Comments for the Author):

Addition of statistical analysis and information of patients improved the manuscript. It will be better if changes of figures and tables are clearly shown.

My minor concern is the following.

ACE2-blocking assay was performed using serial serum dilution, but values are less than 10. This is unusual for me to use such a high concentration of serum. What dilution is used as maximum concentration? It is not clear what "original concentration" means.

Preparing Revision Guidelines

- point-by-point responses to the issues I raised in your cover letter
- Upload a compare copy of the manuscript (without figures) as a "Marked-Up Manuscript" file.
- Each figure must be uploaded as a separate file, and any multipanel figures must be assembled into one file.
- Manuscript: A .DOC version of the revised manuscript
- Figures: Editable, high-resolution, individual figure files are required at revision, TIFF or EPS files are preferred

Please return the manuscript within 60 days; if you cannot complete the modification within this time period, please contact me. If you do not wish to modify the manuscript and prefer to submit it to another journal, please notify me of your decision immediately so that the manuscript may be formally withdrawn from consideration by Microbiology Spectrum.

Dear Editor and Reviewers:

We thank you again for your valuable comments and suggestions. We have revised the manuscript accordingly; our responses to the comments are provided below.

Reviewer #2 (Comments for the Author):

Addition of statistical analysis and information of patients improved the manuscript. It will be better if changes of figures and tables are clearly shown.

- In fig 1, (A) Y-axis labeling "binding activity" replaced the original serology activity, and "(B) Spike- and RBD-binding serological activities during the course of COVID-19 in 25 patients. There were 5, 11, 12 and 20 serum samples collected in the 1st, 2nd, 3rd, and over 4th week of illness, respectively. The mean \pm standard error of mean of days after onset for each group was indicated. The significance between serological activities at different timepoints was determined using one-way ANOVA with post hoc Tukey's test. * $p < 0.05$; ** $p < 0.01$; ns, not significant." was added in fig and legend.
- In fig 2, (B) (C) Y-axis labeling "MBC frequency (spike-specific, %)" replaced the original antispikes MBC activity, and in (A) we described the characteristics of donors: Donor 1 (day 27): a 55-y-o woman with severe COVID-19 with pneumonia. Donor 6 (day 21): a 43-y-o man with moderate COVID-19 and pneumonia. Donor 8 (day 18): a 22-y-o woman with mild COVID-19 and running nose.
-

My minor concern is the following.

ACE2-blocking assay was performed using serial serum dilution, but values are less than 10. This is unusual for me to use such a high concentration of serum. What dilution is used as maximum concentration? It is not clear what "original concentration" means.

- In the study, two-fold serial dilutions of original sera were prepared and tested in the ACE2-blocking assay. The original, undiluted serum that failed to inhibit ACE2-RBD interaction was scored as negative. We have further clarified the method in the revised manuscript (pages 8, 30)
- A flow cytometry-based ACE2-blocking assay using MDCK-ACE2 cells and biotinylated RBD was used to examine the serum activity in the study. Similar assay and reagents have been used to test human/animal serum and monoclonal antibodies in the previous studies (refs 6, 19). Prior to the assay, an optimal amount of biotinylated RBD for producing reliable binding signal was acquired. In the experiment, specific RBD-ACE2 binding can be blocked by COVID-19 patient sera in a dose-dependent manner, but not by sera from healthy controls. The ACE2-blocking titer will be expressed as the reciprocal of the serum dilution presenting 50% inhibition of signal compared to the maximum signal (PBS-biotinylated RBD mixture).

May 18, 2022

Dr. Shu-Hsing Cheng
Taoyuan General Hospital, Ministry of Health Welfare, Taoyuan, Taiwan; School of Public Health, College of Public Health and Nutrition, Taipei Medical University, Taipei, Taiwan
Department of Infectious Diseases
1492 Jhongshan Rd
Taoyuan
Taiwan

Re: Spectrum00743-22R2 (Anti-spike antibody response to natural infection with SARS-CoV-2 and its activity against emerging variants)

Dear Dr. Shu-Hsing Cheng:

Your manuscript has been accepted, and I am forwarding it to the ASM Journals Department for publication. You will be notified when your proofs are ready to be viewed.

Sincerely,

Takamasa Ueno
Editor, Microbiology Spectrum
